# Risk Assessment of Coffee Cherry (Cascara) Fruit Products for Flour Replacement and Other Alternative Food Uses

**DOI:** 10.3390/molecules27238435

**Published:** 2022-12-02

**Authors:** Sara Eckhardt, Heike Franke, Steffen Schwarz, Dirk W. Lachenmeier

**Affiliations:** 1Postgraduate Study of Toxicology and Environmental Protection, Rudolf-Boehm-Institut für Pharmakologie und Toxikologie, Universität Leipzig, Härtelstraße 16-18, 04107 Leipzig, Germany; 2Chemisches und Veterinäruntersuchungsamt (CVUA) Karlsruhe, Weissenburger Strasse 3, 76187 Karlsruhe, Germany; 3Coffee Consulate, Hans-Thoma-Strasse 20, 68163 Mannheim, Germany

**Keywords:** coffee cherry pulp, cascara, novel food, coffee by-products, coffee bean processing, *Coffea canephora*, *Coffea arabica*, caffeine, epigallocatechin gallate, trigonelline, flour replacement, bakery products, risk assessment

## Abstract

Coffee bean harvesting incurs various by-products known for their long traditional use. However, they often still end up being a waste instead of being used to their full potential. On the European market, coffee cherry (cascara) products are not yet common, and a novel food approval for beverages made from coffee cherry pulp was issued only recently. In this article, exposure and risk assessment of various products such as juice, jam, jelly, puree, and flour made from coffee cherry pulp and husk are reviewed. Since caffeine in particular, as a bioactive ingredient, is considered a limiting factor, safe intake will be derived for different age groups, showing that even adolescents could consume limited quantities without adverse health effects. Moreover, the composition can be influenced by harvesting methods and processing steps. Most interestingly, dried and powdered coffee cherry can substitute the flour in bakery products by up to 15% without losing baking properties and sensory qualities. In particular, this use as a partial flour substitute is a possible approach to counteract rising grain prices, transport costs, and disrupted supply chains, which are caused by the Russia–Ukraine war and changing climatic conditions. Thus, the supply of affordable staple foods could be partially ensured for the inhabitants of countries that depend on imported wheat and cultivate coffee locally by harvesting both beans and by-products.

## 1. Introduction

Cascara (i.e., Spanish: husk) is the term typically used to describe the fruity bittersweet layers with unique flavors that cover the coffee bean. Coffee is widely consumed around the world and ranks among the most traded agricultural products [1]. Despite the traditional use of cascara as a beverage prepared with hot water in some coffee processing cultures such as Ethiopia (hashara) or Yemen (qishr) [2], this application is rather unknown in the Western world [3]. About 30% of the dry matter of the coffee cherry is due to the exocarp and mesocarp, which together are generally referred to as pulp [4]. Dry processing produces about 1 kg of coffee husk for every kg of coffee bean produced [5]. Considering that approximately one-third of the world’s population consumes coffee and it is an important export commodity, coffee by-products offer not only great potential but also a large market [6]. Recently, the way to more sustainability was paved, as dried coffee cherry pulp was granted novel food approval in the EU. However, the approval of coffee cherry pulp has so far been limited to its use in non-alcoholic beverages [7,8]. Commercial EU trade in other processed by-products is hindered by the lack of novel food approvals [9] due to the still missing information and risk assessments on bioactive substances.

The cascara pulp obtained during wet processing after harvesting (Figure 1) still has a high moisture and sugar content and therefore spoils quickly [10]. It requires food-safe working steps and equipment for further processing of the pulp. Until now, coffee by-products have been used as fertilizers [11] or animal feed [12]. However, these by-products are suitable only as an admixture to animal feed and not for exclusive feeding, since bioactive substances such as caffeine, tannins, and polyphenols can have undesirable effects in cattle [13]. Bressani et al. showed that an admixture of more than 10% leads to a reduced feed intake, poorer nutrient usage, and reduced weight gain among animals [14]. The high caffeine and tannin contents also represent anti-nutritional factors for plants, for which reason the use of coffee by-products as fertilizers is limited as well [15]. Furthermore, chlorogenic acid has a negative influence on the germination of plant seeds and plant growth [16]. Nevertheless, a large part of the coffee by-products is still inappropriately disposed of in the environment, where the products contaminate soil and water to a high degree. Due to the rapid onset of the fermentation and degradation processes of the coffee cherry pulp and its high biochemical oxygen demand in water, the pH value decreases [17]. Beyene et al. described the change in measured parameters in river water upstream and downstream of coffee plantations in Ethiopia. In summary, the results show significant water degradation during coffee processing: the pH decreased from 7 to 6.2, the amount of oxygen dissolved in the water was reduced, and the load of organic substances was strongly increased. These changes in water quality, as well as the bioactive substances dissolved from the coffee plant, reduced the biodiversity of macroinvertebrates in the river water [7].

Russia and Ukraine together supply 30% of global wheat exports [18]. Due to the Russian invasion of Ukraine and sanctions against Russia, transportation has been severely impeded. Ninety-five percent of the grain was transported through the now destroyed or blocked ports in the Black Sea [19]. Additionally, there is uncertainty about future cultivations and harvests regarding the war situation and drought. In March 2022, the price of wheat increased by 19.7% [20]. One indicator that shows social unrest because of rising food prices is the proportion of income spent on food. In Kenya and Nigeria, for example, it is more than 50% [21]. Especially the MENA (Middle East and North Africa) and Sub-Saharan countries are strongly affected due to their dependence on grain imports from Russia and Ukraine coupled with grain-based nutrition, local political instability, and malnutrition [21]. However, Nicaragua also sources 89% of its wheat from Russia and Ukraine [22]. Somalia and Benin even import all their wheat from Russia and Ukraine [23]. These countries produce coffee themselves or their neighboring countries do. Bentley et al. suggest reducing the usage of and dependence on imported wheat and point to the potential of flour blends of nutrient-rich other crops [20].

The goal of this paper, in addition to presenting the potential of cascara in food production, is to derive safe intake levels and evaluate whether there is a practical benefit for the respective food products reviewed (juice, jam, jelly, puree, flour). In particular, the focus will be on flour and the possibility of partially substituting wheat flour in bread with coffee cherry husk powder.

## 2. Materials and Methods

For the literature search of this review, Google Scholar and PubMed databases, as well as the European Food Safety Authority (EFSA) homepage and official EU regulations were used. The search terms used included coffee cherry pulp, coffee cherry husk, caffeine, trigonelline, and epigallocatechin gallate (EGCG). The identified references were obtained in full text and combined into a narrative review.

## 3. Coffee Cherry Structure and Processing

In the literature, the reported contents of components vary significantly in some cases. The reason for this situation, apart from geographical and biological conditions, is also the different understanding of what cascara is. Depending on the processing method, different parts of the fruit are included in the final product. The processing methods and the structure of the coffee cherry will be briefly outlined in the following.

### 3.1. Coffee Cherry Structure

Coffee cherries are drupes that grow to an average diameter of 1.5 cm (Figure 2). The two main species that are grown are *Coffea canephora* and *Coffea arabica* [11]. Between the parchment and the outer skin lies the sweet-tasting mesocarp. The two coffee beans themselves are coated with silver skin and parchment (Figure 3).

### 3.2. Coffee Cherry Processing

When processing, a distinction is made between the dry method, the wet method, and the semi-dry or semi-washed method (Figure 4).

The dry method is often used to process *C. canephora,* but also for *C. arabica* (as the main processing method in countries such as Brazil or Ethiopia) [24]. During this process, the coffee cherries are spread on fields, terraces, or beams to be dried in the sun (Figure 5), if necessary, with the help of heated air dryers, until the moisture content has been reduced to approximately 10–12% [25]. This takes, depending on the local climate, about 1 to 2 weeks. Subsequently, a dehulling machine separates the green coffee bean from the dried outer layers of the fruit, thus obtaining coffee husks (Figure 6). By definition, coffee husks or cherry husks consist of the exocarp (outer skin), the mesocarp (pulp), and the endocarp (parchment). In the more sophisticated wet method, ripe coffee cherries are separated in flotation tanks, after which a mechanical depulper separates the skin and pulp from the green coffee bean (Figure 7) as well as from the green (unripe) cherries which cannot be depulped. The coffee bean is still covered with parchment and mucilage layer and is now soaked and fermented for 12–36 h. Fermentation is performed using existing or added microorganisms or enzymes [25]. The microbial community during depulping is mainly composed of lactic acid bacteria, acetic acid bacteria, enterobacteria, and yeasts. During fermentation, the proportion of lactic acid bacteria increases, such as *Leuconostoc pseudomesenteroides*, *Lactobacillus vaccinostercus*, *Lactobacillus brevis*, and *Lactobacillus plantarum* [26]. Before finally drying the beans, they must undergo intensive cleaning with water. Coffee beans obtained using this method often have a higher market value due to the higher acidity created during the fermentation process [11,25]. The by-product separated during the depulping process is called coffee pulp or cherry pulp, which consists of exocarp and mesocarp. During the wet method, the parchment remains attached to the green coffee bean and is degraded by fermentation processes.

During the semi-dry method (often also referred to as “pulped natural”), as during the wet method, the ripe coffee cherries are sorted in flotation tanks, and the pulp is removed from the green coffee bean. However, no fermentation occurs after this process; instead, the mucilage remains on the bean and is dried along with it [27]. This method was developed in Brazil [25]. Coffee beans processed this way are claimed to have a particularly flavorful body due to the remaining layers rich in polysaccharides and give the beverage a certain sweetness [28].

## 4. Chemical Characteristics of Coffee Cherry Pulp and Cascara

### 4.1. Caffeine

Caffeine (Figure 8) is a purine alkaloid found in many parts of the coffee tree. In the main biosynthetic pathway, xanthine is first converted to 7-methylxanthine, then to theobromine, and finally to caffeine with three methylations [29]. 

After consuming foods containing caffeine, almost all of the caffeine is absorbed into the stomach and intestine and distributed throughout the body [29]. The inhibitory effect of caffeine on various subtypes of the adenosine receptor is responsible for its effects in the human body. Inhibition of the A1 receptor, which is located primarily in the central and peripheral nervous systems, has been shown in long-term studies in patients and animal models, as well as epidemiologic studies, to have positive effects on various diseases [30]. 

Since large parts of the population consume caffeine daily and chronically in the form of beverages such as tea, coffee, soft drinks, or in the form of chocolate, the next section of this review will calculate the caffeine content of coffee cherry products and develop a risk assessment. A mild overdose of caffeine can manifest itself as sleep problems, mild anxiety, accelerated breathing, increased diuresis, and cardiovascular effects. Higher doses cause tremors, headaches, insomnia, and even delirium [31]. The effect of caffeine is highly inter-individual variable, due to genetic factors and habituation effects. In adults, a single dose intake of 200–300 mg caffeine shows a blood pressure increase of 3–8 mmHg (systolic) and 4–6 mmHg (diastolic). The peak is reached after 60–90 min, but with repeat dose consumption, this effect can last up to 12 h. This observation is independent of age (adults) and sex [32].

Reduced sleep duration and increased sleep latency are possible in children, adolescents, and adults from single doses of 1.4 mg/kg bw, if caffeine is consumed shortly before sleep. From a single dose of 400 mg of caffeine, or when taken in a relatively short time interval, an increase in anxiety is possible in adults. In older studies, it has been shown that even at single doses of 10 mg/kg bw, there is no significant increase in anxiety in prepubertal children and adolescents [33,34]. More recent data on the effects of caffeine on children are rare because studies with children have been avoided. 

The caffeine content obtained in different studies differs considerably. Table 1 summarizes the type of coffee, sample preparation method, and analytical methods used to determine the caffeine content. Additionally, the content data were converted to g/kg of coffee cherry material to provide better comparability.

To obtain a value for risk assessment, the mean value of all caffeine contents listed in Table 1 that were measured in the cherry pulp or husk of *C. arabica* was determined. The average caffeine content is 3.8 g/kg of dried coffee cherry pulp/husk. The median is 3.6 g/kg and the P90 is 6.65 g/kg.

Since there are fewer sources for caffeine amounts in *C. canephora* pulp or husk, the mean value is less informative for risk assessment purposes. Additionally, Thaiphanit et al. are omitted from the mean calculation, as the measured values showed a strong deviation due to the method of analysis [37]. This study is included in Table 1 because the experiment allows important conclusions on how the caffeine content can be influenced. 

The caffeine content of a sample is not only dependent on the type of coffee but can also be influenced by a variety of other factors. Here are several parameters and to what extent they change the caffeine content:Delayed depulping after harvesting causes the initiation of microbial fermentation processes in the coffee cherries. The delay can be produced by dry processing or by storing the coffee cherries in a bag, in water, or at room temperature in a basket for 12 h before depulping. Arpi et al. concluded from the data that microbial caffeine degradation starts after 12 to 36 h [1]. As shown in Table 1, the wet-processed sample had the highest caffeine content at 4.5 g/kg [1]. Therefore, the conclusion can be drawn from this result that wet-processed coffee cherries generally have higher caffeine content. It is also possible that this correlation occurs only when considering one coffee species at a time with its specific growth conditions, but different processing methods.The solvent temperature and the extraction time have a significant influence on the amount of caffeine that will be contained in the cascara-based beverage. Here, the following applies: the hotter the water and the longer it is brewed, the higher the measured caffeine content [37].If the cascara samples are blanched in hot water for 1 min and then dried again, the caffeine content in the subsequently brewed beverage may be up to 40% lower compared to the unblanched cascara sample [39]. Since caffeine is water-soluble, the caffeine content in the raw product can be considerably reduced with a preceding blanching step.Other causes of fluctuations in caffeine content may also be species and variety, the time of harvest, climatic conditions, soil properties, and geographical altitude. For example, Belitz et al. [11] show that roasted coffee beans from *C. canephora* have a caffeine content of 2.4%, which is almost twice that of *C. arabica*.

### 4.2. Epigallocatechin Gallate

Epigallocatechin gallate (EGCG) is probably the best-known antioxidant in green tea and, due to its structure, is classified as polyphenol and catechin. EGCG is an ester of gallic acid and the catechin epigallocatechin (Figure 9). There are numerous pharmacological and clinical studies referring to this polyphenol. Undoubtedly, many positive effects on human health can be attributed to EGCG, such as its antioxidant and anti-inflammatory effects or the reduction of blood pressure and blood cholesterol levels [41].

Green tea extracts containing high contents of EGCG are also marketed as dietary supplements, with some cases of possible hepatotoxicity having been reported [42]. Mazzanti et al. [42] analyzed 34 cases of hepatitis in the context of EGCG in the period 1999 to 2008. There is a large increase in liver enzymes, such as transaminases, alkaline phosphatases, and gamma-glutamyl transpeptidases. As a result, the bilirubin level also shows a strong increase. Additionally, Cai et al. [43] could show in studies with mice that high doses of EGCG (500 and 1000 mg/kg/day) trigger cardiac fibrosis by downregulating the phosphorylation of AMPK and subsequent signaling pathways. However, such doses are difficult to achieve only by consuming natural drinks containing EGCG. Much more hazardous are dietary supplements made from the respective extracts [43]. Therefore in 2015, EFSA initiated a risk assessment to derive safe intake levels [44]. The EFSA Scientific Opinion published in 2018 states that the average daily intake of epigallocatechin gallate by the population ranges from 90 to 300 mg. Furthermore, EFSA states that there is no evidence of hepatotoxicity when up to 800 mg of EGCG is consumed daily over a period of 12 months [44]. 

This review provides a brief risk assessment of EGCG in coffee cherry pulp and cherry husk products, but it should be noted that, although traces of the considered polyphenol are occasionally found in coffee cherry, it plays a rather negligible role in terms of quantity. Moreover, there are very few references in the literature, which exclusively indicate the EGCG content. Sangta et al. [45] state that they processed Arabica coffee cherries pulp themselves into powder and extracted it with methanol. A flavonoid content of 1.03 µg/g of dried material and a polyphenol content of 0.56 µg/g were measured. The authors concluded from further experiments that EGCG makes up approximately one-third of the fraction of flavonoids and polyphenols [37]. Evidently, the EGCG content is negligibly small and often below the detection limit.

### 4.3. Trigonelline

Trigonelline is a pyridine alkaloid (Figure 10). It differs from nicotinic acid, better known as vitamin B3, only by an additional methyl group at the nitrogen atom [46]. It is soluble in water [47], implying that it is also detectable in beverages made from coffee cherries. Hu et al. stated that the content of trigonelline is usually higher in the pericarp than in the coffee beans themselves. In some cases, this balance is reversed when redistribution occurs in the coffee cherry [29]. 

Trigonelline performs a wide variety of tasks in plants, such as leaf movement [48], acts as an osmolyte, provides adaptation to stress situations [49] and regulates the cell cycle [50]. Generally, trigonelline accumulation is observed during stress situations or under special growth conditions [51]. During the past two decades, numerous findings have been made on the physiological and pharmacological effects of trigonelline in humans. Tohda et al. described neuroprotective effects when observing the regeneration of dendrites and axons in vitro [52]. Additionally, trigonelline is also capable of reducing human insulin levels in vivo, thus showing antidiabetic effects [53]. 

Despite observations that trigonelline shows in vitro anticarcinogenic potential through induced apoptosis, suppression of nuclear factor erythroid 2-related factor 2 (Nrf2) activity, induction of cytoprotective genes [54], as well as inhibition of reactive oxygen species (ROS)-induced proliferation of invasive cells [55], a correlation has also been found in the development of breast cancer. Trigonelline is now considered a phytoestrogen [56]. In particular, trigonelline is structurally very different from classical phytoestrogens, such as genistein, which is more similar to estradiol. Therefore, it is not surprising that trigonelline does not exert its effect via ligand–receptor bonding [56]. Allred et al. [56] revealed that trigonelline increases the expression of estrogen receptor target genes in MCF-7 cells (a breast cancer cell line) to the same extent as estradiol. Consequently, it stimulates the growth of estrogen-dependent breast cancer cells. Additionally, Allred et al. [56] found in further experiments that MCF-7 cell growth is dose-dependent. The concentration required for this is reported to be between 10 and 100 pmol/L. 

In the literature, there is only limited data on the trigonelline content in coffee cherry pulp or husk. Mostly, the content is given in relation to the green or roasted bean. Cangussu et al. [39] determined the content in various coffee cherry husk samples (Table 2).

For risk assessment, the value of the unblanched pulp with low parchment content is used, as this corresponds to the material used most frequently.

The content of trigonelline in the final product depends on the following variables and can be partly influenced by processing steps:As observed in the experiment by Cangussu et al. [39], the trigonelline content in coffee cherry husks can be almost halved if they are pretreated with a blanching step. The reason for this is the water solubility of trigonelline.The measured values indicate that more trigonelline is enriched in the coffee cherry pulp than in the parchment.There are definitely species-specific variations in trigonelline content. The contents are usually only indicated for coffee beans, and studies on the chemical composition of coffee cherry components are still needed. It can be seen that the trigonelline content in *C. arabica* beans is higher than in those of *C. canephora* [57]. Additionally, the trigonelline content decreases during roasting [58]. A transfer of the data to the coffee cherry pulp and husk would be inappropriate, since an inverse relationship is already known for the caffeine content: coffee beans of *C. canephora* contain more caffeine than *C. arabica* [11], while the coffee cherry pulp of *C. arabica* contains more caffeine than *C. canephora* [37].The content of trigonelline increases with the geographical altitude. This was measured in green coffee beans from *C. canephora* plants cultivated at different altitudes [51]. The difference is up to 25% more trigonelline in a plant grown at a higher altitude compared to one grown at a medium to low altitude. This is probably related to the function of trigonelline in adapting the coffee plant to stress situations.

## 5. Coffee Cherry Food Products

### 5.1. Juice

The literature often refers to cascara as a beverage, which is generally understood as the infusion of dried coffee cherry husk or pulp with hot water to obtain a tea-like beverage. Heeger et al. [59] describe the production of a cascara beverage in which 65.6 g of dried coffee cherry pulp is brewed for 6.5 min with 1 L of 90 °C hot water. The beverage was then mixed with 5.7 mL of lemon juice and 71 g of sugar and pasteurized after bottling. Thus, a drink very similar to iced tea is obtained. However, it is less common to actually find the production of juice from coffee cherry husk and/or pulp that corresponds to any definition of the EU regulation. The work of Velissariou et al. [60] describes the production of a beverage in which an extract of coffee cherry husks and pulp is first obtained, and subsequently, various beverage formulations can be developed from it. The production process meets the definition “fruit juice from concentrate” from Annex I of the Council Directive 2001/112/EC: “The product obtained by replacing in the concentrated fruit juice water extracted from that juice during concentration, and restoring the flavors, and, if appropriate, pulp and cells lost from the juice but recovered during the process of producing the fruit juice in question or of fruit juice of the same kind” [61].

In the production process from Velissariou et al. [60], dried coffee cherry husk (*C. arabica* var. Mundo Novo) is used. In example 1, 500 kg of dry matter is mixed with 2500 kg of water, and 1300 kg of extract is obtained at the end. This results in a multiplier of 2.6 from dry matter to extract. Furthermore, the beverage/juice should contain between 5 and 30% extract. Assuming the maximum planned amount of 30% extract in the final drink, this corresponds to approximately 300 g of extract per liter of water (ρ = 997 g/L). To obtain the mass of dried raw material, 300 g must be divided by 2.6, and 115.38 g of dry matter used per liter of juice is obtained. With 5%, this corresponds to 19.23 g of raw material.

Velassariou et al. state that the extract should be standardized so that a caffeine content of 0.07 mg/mL to 0.30 mg/mL can be adjusted in the finally obtained beverage [52]. Consequently, 1 L of fruit extract juice contains between 70 and 300 mg of caffeine. 

Regarding the content of trigonelline, the amount of dried raw material used for 1 L of juice was considered. With the trigonelline value of 5.4 g/kg determined by Cangussu et al. for unblanched coffee cherry pulp with parchment [39], a trigonelline content of about 0.1–0.6 g/L for 5–30% extract was obtained.

### 5.2. Jam

Another possibility of using coffee cherry pulp could be jam production. So far, no sources can be found in the literature. The official definition of jam in Annex I of Council Directive 2001/113/EC is as follows: “‘Jam’ is a mixture, brought to a suitable gelled consistency, of sugars, the pulp and/or purée of one or more kinds of fruit and water. … The quantity of pulp and/or purée used for the manufacture of 1000 g of finished product must not be less than: 350 g as a general rule“ [62].

According to this definition, a hypothetical recipe that complies with the EU requirements for jams was developed to derive the content of bioactive substances for risk assessment. To make 1000 g of jam, 350 g of fresh coffee cherry pulp is needed. The fresh material had a moisture content of about 80% [63]. Assuming classical sun drying followed by hot air drying, a moisture content of 10–12% results [25]. For simplification, a drying loss of 70% can be estimated. For 350 g of fresh coffee cherry pulp, this is equivalent to 105 g of dried material. This derivation is necessary because, in the literature, the data for caffeine and trigonelline are related to dried coffee cherry pulp. For caffeine, the mean value of 3.8 g/kg as given in Section 4.1 can be used. Consequently, 1 kg of jam can contain about 399 mg of caffeine. A commercially available jar of jam in Germany usually contains 450 g. This corresponds to a total amount of about 180 mg of caffeine. An average serving of jam is 20 g, which corresponds to approximately 1 tablespoon and therefore contains approximately 8 mg of caffeine.

If 1 g of dried coffee cherry husk contains approximately 5.4 mg of trigonelline, then a 450 g jar of jam contains 255 mg, and a tablespoon of 20 g contains 11 mg of trigonelline (Table 3).

### 5.3. Jelly

“‘Jelly’ is an appropriately gelled mixture of sugars and the juice and/or aqueous extracts of one or more kinds of fruit. The quantity of juice and/or aqueous extracts used in the manufacture of 1000 g of finished product must not be less than that laid down for the manufacture of jam. These quantities are calculated after deduction of the weight of water used in preparing the aqueous extracts”.[54]

According to the Council Directive definition, a similar amount (105 g) of dry material is used to produce 1000 g of jam. The difference is that jelly is made from juice, to which sugar, pectin, and acid are added to enable gelation. Therefore, the caffeine and trigonelline contents correspond to the values calculated for jams. Devoney [10] presents an impressive attempt to develop a simple process for jelly production to be used by local coffee farmers. The procedure does not require extraordinary and expensive equipment, and the recipes are understandable and easy to implement. Coffee farmers must have access to clean water, a source of heating, and require lemons for the independent production of pectin and sugar for preservation. The coffee cherries need to be heated and pressed soon after depulping. The spent cascara can be fed to ruminants because boiling extracts the anti-nutritional factors. 

A sensory survey was also conducted, and the main flavors were determined, as well as the reaction of the taste testers to different recipes. Overall, the jelly was rated positively; the few negative ratings related to the strong sweetness or the fermented taste with a slightly bitter note in the aftertaste. This points to the difficulty that the fresh coffee cherry pulp must be processed directly after harvesting, as fermentation processes start quickly. The dominant flavors are fruity, reminding of plum and fig, and sweet, similar to honey, caramel, or brown sugar. Additionally, earthy and floral nuances are also described. It is worth mentioning that Devoney [10] also makes a caffeine calculation analogous to the way in this study—find a caffeine value for coffee cherry pulp from the literature and calculate the drying loss as the mass difference between fresh and dried material. However, a much higher value was obtained than in the estimation in this study: 700 mg of caffeine for a glass of jelly and 44 mg of caffeine for a tablespoon of about 14 g. The reason for this result is that the considered jelly recipe uses 500 g of fresh coffee cherry pulp for a 227 g jar, which corresponds to a much higher amount of starting material than that required by the EU regulation. The effect of jelly formulations with lower fruit content on sensory characteristics remains to be investigated.

### 5.4. Puree

The process of producing purees is illustrated by Buck et al. [67]. A distinction is made between processed puree and unprocessed puree. For this purpose, freshly frozen whole coffee cherries from *C. canephora* are used. After defrosting and brief blanching, unripe or shriveled berries are sorted, and green coffee beans are removed by hand, analogous to wet processing. For the processed puree, 150 g of water, 2.5 g of citric acid, and 47.5 g of sucrose were added to 100 g of fresh coffee cherry pulp, which yielded 300 g of puree. Again, assuming a 70% difference in moisture content between fresh and dry coffee cherry pulp, 100 g of fresh pulp corresponds to 30 g of dried material. 

Consequently, according to the values determined above, 300 g of processed puree contains approximately 120 mg of caffeine and 162 mg of trigonelline. EGCG is neglected, because the value is too low. Calculated on 100 g of puree, for clear comparability of both types of puree, this amounts to 40 mg caffeine and 54 mg trigonelline.

For the unprocessed puree, the coffee cherry pulp is pureed only with water and contains no other additives. This implies that for every 100 g of fresh material, 150 g of water is used. An amount of 250 g of puree contains 120 mg of caffeine and 162 mg of trigonelline. Calculated on 100 g of puree, this corresponds to 48 mg of caffeine and 65 mg of trigonelline. 

There are some advantages of processed puree when compared to the unprocessed one: the addition of citric acid prevents the enzymatic and nonenzymatic degradation of phenolic ingredients such as anthocyanins. This is reflected in the stable red color of the puree after production and the antioxidant capacity of the processed puree, which is significantly higher than that without added citric acid [67,68].

Sensory evaluation of the processed puree by Buck et al. [67] revealed that it was mostly described as sweet, followed by sour, bitter, and astringent. The retronasal evaluation put fruity, apple-like notes in the foreground, after which the testers also described floral, hay-like, citrus-like, musty, and earthy facets.

### 5.5. Powder/flour

The use of coffee cherry pulp or husk as flour is possible up to a certain amount. In their review, Iriondo-DeHond et al. [69] stated that in gluten-free bread 3–4.5% can be replaced by isolated coffee cascara dietary fiber and 2.5% can be replaced by cascara extract without a negative influence on the baking procedure, and the sensory result. Additionally, the bread has a higher crumb elasticity, and the color of the cascara gives the bread the appearance of wholewheat bread. It should be noted that the description of Iriondo-DeHond et al. [69] does not refer to cascara flour, but to coffee cherry pulp processed differently. A more detailed description of the experiment and the nutritional analysis of coffee cherry husk extract can be found in Guglielmetti et al. [70]. The process steps by which dietary fiber is obtained and integrated into gluten-free bread recipes are described in more depth by Rios et al. [71]. 

The production of conventional white bread with coffee cherry powder was described by Rosas-Sanchez et al. [72]. They even use decaffeinated pulp powder and substitute wheat flour in mass proportions of 1.25%, 2.5%, and 5% (wet basis). With more than 5% powder, the bread gets an earthy aftertaste, with more than 10%, the dough becomes difficult to handle. In contrast to gluten-free bread, regular wheat bread has a harder crust and a higher proportion of pulp powder. In general, the bread becomes darker and denser, but consumers accept a substitution of up to 5% in terms of taste. The caffeine content of the decaffeinated coffee powder is indicated as 0.04 g/100 g. This corresponds to 0.4 g/kg of coffee cherry powder and is therefore negligibly low [72].

The baking experiment of Serag El-Din et al. [73] on balady bread is particularly informative. The balady bread is the national Egyptian bread, which is similar in style to flatbread. The traditional recipe consists of 100 g of wheat flour, 0.5 g of yeast, 1.5 g of cooking salt, and about 80 g of water. In the experimental series, 5%, 10%, 15%, and 20% of wheat flour were replaced with coffee cherry husk powder, respectively. In addition to sensory analysis by 15 participants, the impact of the bread on the blood lipid levels of hyperlipidemic rats was investigated.

The sensory evaluation shows that with a substitution of 5–10% of the wheat flour, a flatbread can be baked that is still very much comparable to the original. With 15% coffee cherry husk powder, the balady bread is still rated good overall. With 20%, there was a sharp decline in the positive evaluation of appearance, flavor, and taste. Color and crumbs are also no longer considered particularly appealing. Concerning the baking properties, it should be mentioned that dough density increases with increasing coffee cherry husk content, while volume and height decrease accordingly. The flatbread comparably shows the same cohesive behavior, but gumminess, chewiness, and firmness are strongly reduced. 

The subsequent bioassay shows that the proportion of coffee husk powder in the balady bread shows an inverse relationship with triglycerides, LDL cholesterol, VLDL cholesterol, and lipid values in the rat blood. The atherogenic index also decreases. The HDL cholesterol showed a slight increase. If the results can be transferred to humans in further studies, the use of coffee cherry husk powder as a partial flour substitute was hypothesized to have possible beneficial health effects. 

Serag El-Din et al. determined a caffeine value for coffee cherry husk powder, which is comparatively high at 18.2 g/kg of dry material. If calculated with a value of 18.2 g of caffeine per kg of dry material determined by the authors and a flatbread made from a 125 g dough piece, caffeine values of 63 mg, 125 mg, and 188 mg are obtained for 5%, 10%, and 15% coffee cherry husk content, respectively. If the estimated mean value of 4.0 g/kg of this study is used, 14 mg, 28 mg, and 41 mg of caffeine in a flatbread with 5%, 10%, and 15% flour substitutes are obtained, respectively. The trigonelline content in a flatbread for 5, 10, and 15% amounts to 19 mg, 37 mg, and 56 mg for the estimated value in this study.

Coffee cherry pulp flour may not have the same baking characteristics as wheat flour, but with 62% it has a comparable carbohydrate content to wheat with 69% [74]. The protein content ranges from 8 to 12% for wheat flour and 10% for cascara. Cascara can add particular nutritional value to the baked product with its relatively high dietary fiber content of 21% (wheat flour: 1–2%) and the vitamin C it contains. It is discussed whether the fiber-related α-amylase inhibition lowers glucose production from starch and also decreases starch digestibility. Cascara flour, with its antioxidant properties and its ability to retard glucose and lipid absorption, could have a positive impact on health [75]. Wiliana et al. summarized the advantages of cascara flour over other flours and flour substitutes in its availability (as a harvest by-product) and its nutritionally better values than, for example, corn or rice flour [74].

Coffee cherry pulp (flour) in addition has an iron content worth mentioning. The highest content measured in a sample by Setyobudi et al. is 259 mg Fe^3+^/kg. Although this is trivalent non-heme iron, the contained vitamin C, beta-carotene, and reducing sugars enhance iron bioavailability [76]. As some African countries are dependent on grain trade with Russia and Ukraine, the possibility of substituting up to 15% of wheat flour in such an easy-to-implement flatbread recipe with little technical effort offers interesting possibilities. Nevertheless, the flour production process and the cost of using it as a flour substitute need further investigation and could vary greatly depending on the country of origin and destination. On the other hand, it does not involve additional costs for cultivation, sorting, and harvesting [13].

## 6. Exposure Assessment

For the exposure assessment, the assessment of caffeine, epigallocatechin gallate, and trigonelline is combined with intake data from the German population. The German National Nutrition Survey II (NVS II) is an extended epidemiological study in which 20,000 German-speaking inhabitants of Germany between 14 and 80 years of age were interviewed about their dietary habits and food consumption in 2005–2007 [77].

The NVS II [78] compiled lists of average values for the daily intake of various foods by age group and sex. 

EFSA also uses data from the VELS (“Verzehrsstudie zur Ermittlung der Lebensmittelaufnahme von Säuglingen und Kleinkindern für die Abschätzung eines akuten Toxizitätsrisikos durch Rückstände von Pflanzenschutzmitteln”—English: consumption study to determine the food intake of infants and young children for assessing acute toxicity risk from pesticide residues) study conducted in 2000, which collected the statistical distribution of food intake amounts of infants and toddlers [79].

### 6.1. Exposure to Caffeine

Since caffeine clearance in children and adolescents is comparable to that of adults [32], with a realistic risk assessment, it can be deduced that for children aged 3 years and older, as for adults, 3 mg of caffeine per kg of body weight (bw) as a single dose intake is not associated with an adverse health risk. This value also applies to lactating mothers. Pregnant women should not consume more than 200 mg of caffeine in total per day. Otherwise, there may be a risk of fetal hypoxia due to uteroplacental vasoconstriction due to the easy plantar passage of caffeine. In addition, caffeine consumption during pregnancy may be associated with reduced birth weight [32]. Table 4 summarizes the data evaluated by the EFSA in the Scientific Opinion on Caffeine.

The highest average consumption value of juice among adolescent males between 14 and 18 years is 460 g per day [78]. Body weight in this age group varies widely. The literature data states 57 kg (14 years) to 77 kg (18 years) for Germany [80]. If we assume the worst possible case that a 14-year-old male adolescent drinks a coffee cherry pulp juice with a particularly high caffeine content of 300 mg/L, we obtain 573 mL for a single dose intake. For juices with a low content of 70 mg/L, it is already more than 2.4 L that can be drunk at once without negative adverse effects. The possible daily intake is accordingly almost twice as high (see Table 4). With appropriate warnings about the caffeine content on the label, it seems to be acceptable for adolescents over the age of 14 to drink juice made from coffee cherry pulp.

The amount of jam and jelly, summarized as sweet spreads, coincides well with the portion of 20 g often indicated on the outer packaging of food products. This also corresponds to the assumption of a tablespoon. If we again take as an example the same German male 14-year-old adolescent with an average weight of 57 kg, who, according to NVS II, consumes 20 g of jam daily, this results in a value of 430 g of jam that can be safely consumed in one session. Even with natural variations in ingredient levels and different weights of consumers, adverse effects from jam consumption are a rather unrealistic scenario.

The NVS II does not provide any precise information on the average intake amounts of puree or fruit puree.

If we consider the average intake of bread of the particularly vulnerable group of 14- to 18-year-olds, it is 142 g per day for girls and 182 g per day for boys. The average consumption of older people in Germany ranges from 118 to 184 g per day. The amount of 184 g of bread with an admixture of 2.5% cascara flour contains 18.4 mg of caffeine. This is also very low if the safe single dose values from Table 4 are considered.

On average, humans up to 18 years of age consume a maximum of 1.0 mg/kg bw daily, although up to 5.7 mg/kg bw would be safe. This suggests that moderate consumption of coffee cherry products, such as jam or juice, does not lead to exceeding limit values, even in children and young people. The risk assessment is based on single dose values, since food intake is a temporally limited event in daily routine.

### 6.2. Exposure to Epigallocatechin Gallate

The literature research shows that EGCG plays a minor role in the analysis of the ingredients in the coffee cherry pulp. The content was determined only in a paper by Sangta et al. [45] and lies in the µg range. Even with excessive consumption of foods made from coffee cherry pulp or husk, it is not possible to exceed the limit of 800 mg/day determined by EFSA to be safe for adults [44]. Therefore, there is no risk of adverse effects, such as liver damage, of EGCG when consuming products containing or made from coffee cherry pulp or husk.

### 6.3. Exposure to Trigonelline

Recently, trigonelline has been described and studied in the context of its beneficial properties for human health, rather than in the context of its toxicity. In the literature, no limit values (acceptable daily intake) are proposed for trigonelline; but only values from older animal experiments were available. In their review, Zeiger et al. [81] summarized the toxicological data collected on trigonelline. In rats, the oral and subcutaneous LD_50_ values were 5000 mg/kg. By adding 50 mg/kg of trigonelline to the diet of mice daily for 21 days, no change in organ weights can be determined, including the uterus and ovaries. No visible effects can be observed in cats consuming 3500 mg of trigonelline by food daily for 62 to 70 days. Looking at the values obtained in subacute animal experimental studies, where no adverse effects could be observed, a range that is far from even excessive human consumption is reached. For example, in a cup of espresso (V = 25 mL), up to 72.7 mg of trigonelline may be present, depending on the preparation method and the type of coffee [82]. However, only a vague estimate remains possible, as well as the necessity of collecting toxicological data in order to provide a reasonable risk assessment. Therefore, the safe intake amounts from Table 5 are derived exclusively from the caffeine content. However, trigonelline contents are reported and can be used for risk assessment when better data for this compound become available in the future. 

## 7. Conclusions

The potential of the coffee cherry pulp and husk is obviously still not fully exploited. Currently, only the marketing of dried pulp of coffee cherry and the infusion thereof is authorized in Europe by EU Regulation No. 2022/47 of 13 January 2022.

With the abundance of potential foods, there is the possibility of obtaining further novel food approvals in Europe. This not only means a decrease in environmental pollution and a step toward increased sustainability, but also results in a commercial expansion of trade for coffee farmers.

There are hardly any critical ingredients, and the risk assessment shows that the safe intake amounts for the evaluated coffee cherry pulp and husk products even exceed the average amounts consumed by the population. The data also show that moderate consumption of pulp and husk foods is not associated with adverse health risks even for younger age groups but should be excluded for safety reasons for pregnant women as well as nursing mothers and provided with an appropriate warning on the product label.

Furthermore, the literature research shows that up to 2.5% of wheat flour can be replaced in bread and up to 15% of it in flatbread. This opens up new possibilities in times of rising energy prices, which in turn lead to increasing costs of land cultivation and irrigation in the agricultural sector. The analysis also shows that the Russian invasion will affect the food security of countries with limited coping capacity. These include mainly MENA (Middle East and North Africa) and Sub-Saharan countries, which are highly dependent on grain imports from Russia and Ukraine, have grain-based diets, partial political instability, and undernourishment. Yemen, Egypt, Tunisia, Lebanon, Congo, and Tanzania are particularly affected. These countries or their neighboring countries produce coffee themselves. Since coffee cherry husk is discarded as waste anyway, processing it into flour is an interesting approach to consider. It could be used locally for partial substitution of the currently expensive imported flour.

Finally, it can be said that even if the bioactive ingredients might limit the application, there are possibilities to reduce these substances. For example, Rosas-Sánchez et al. proposed the use of decaffeinated cascara flour in bread production [63]. Alternatively, there are methods to influence the caffeine and trigonelline content, such as blanching. Last but not the least, the choice of coffee species also has an essential influence on the ingredient profile of the product. 

## Figures and Tables

**Figure 1 molecules-27-08435-f001:**
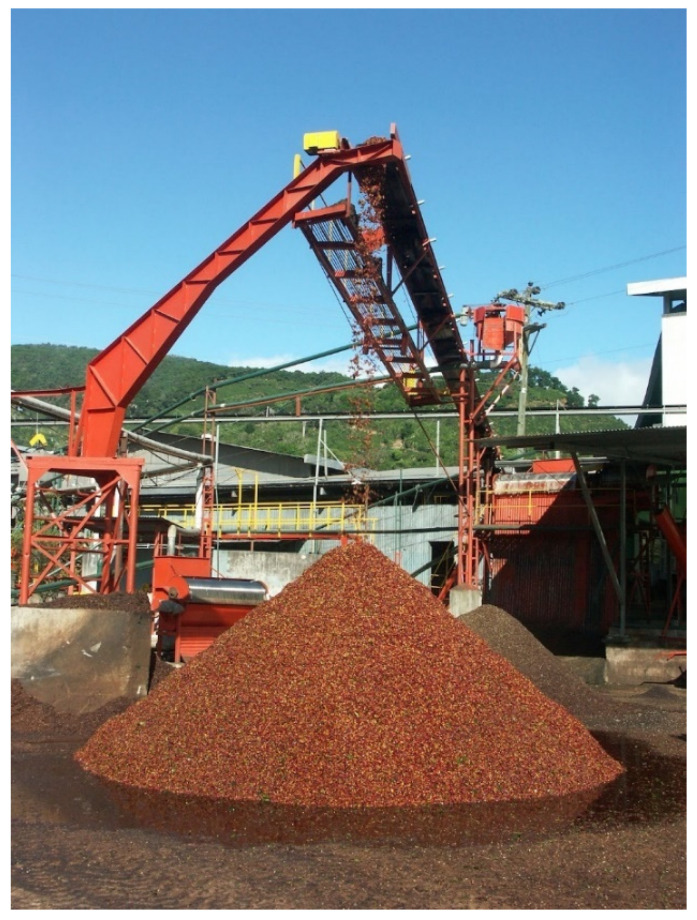
Coffee pulp from wet processing collected in a heap.

**Figure 2 molecules-27-08435-f002:**
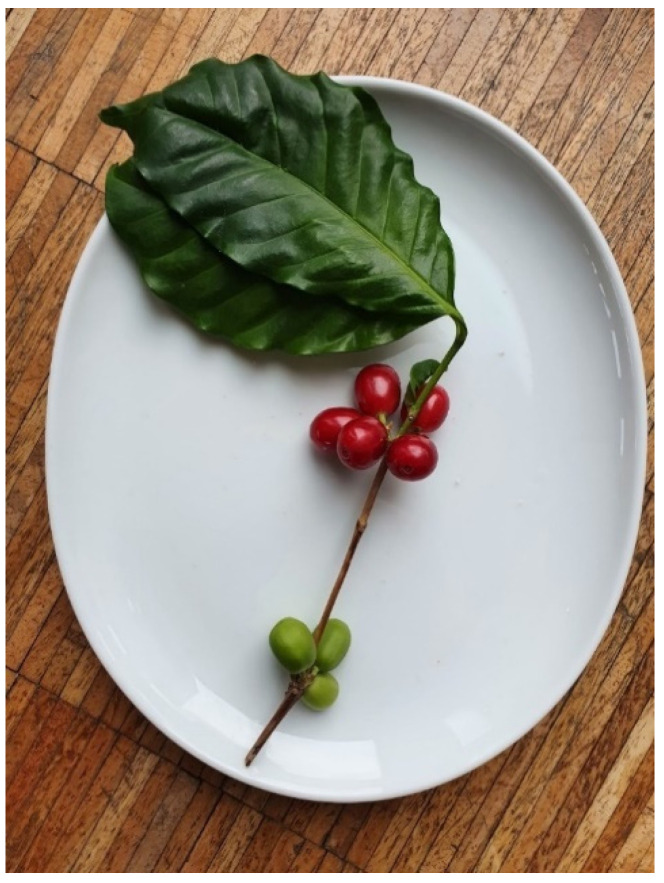
Ripe (red) and unripe (green) coffee cherries on a branch.

**Figure 3 molecules-27-08435-f003:**
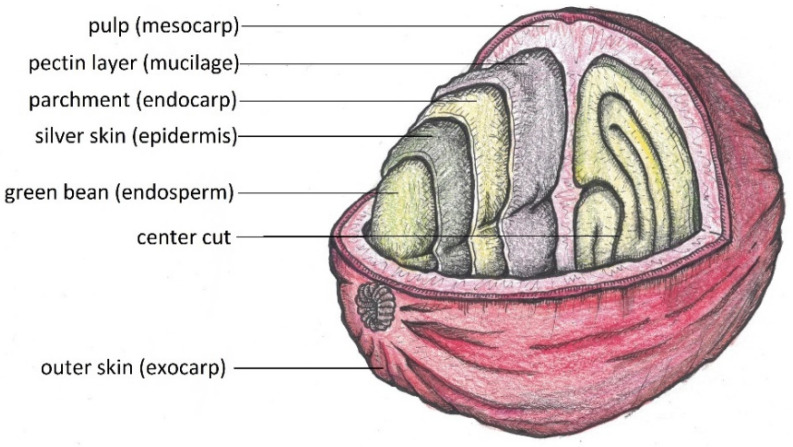
Different layers of a coffee cherry (drawing by the authors).

**Figure 4 molecules-27-08435-f004:**
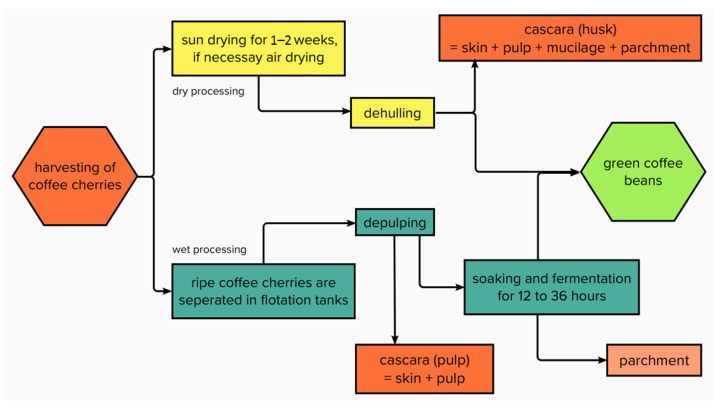
Dry and wet processing of coffee cherries.

**Figure 5 molecules-27-08435-f005:**
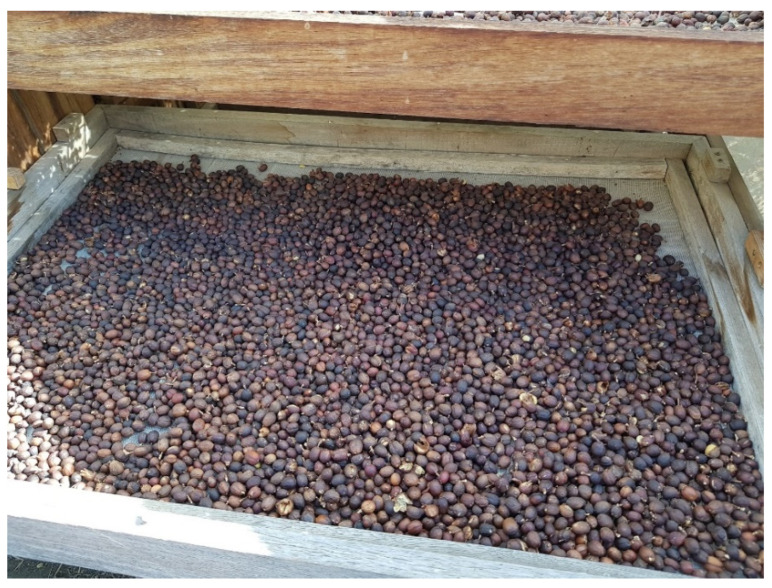
Coffee cherries are spread on beams to be dried in the sun.

**Figure 6 molecules-27-08435-f006:**
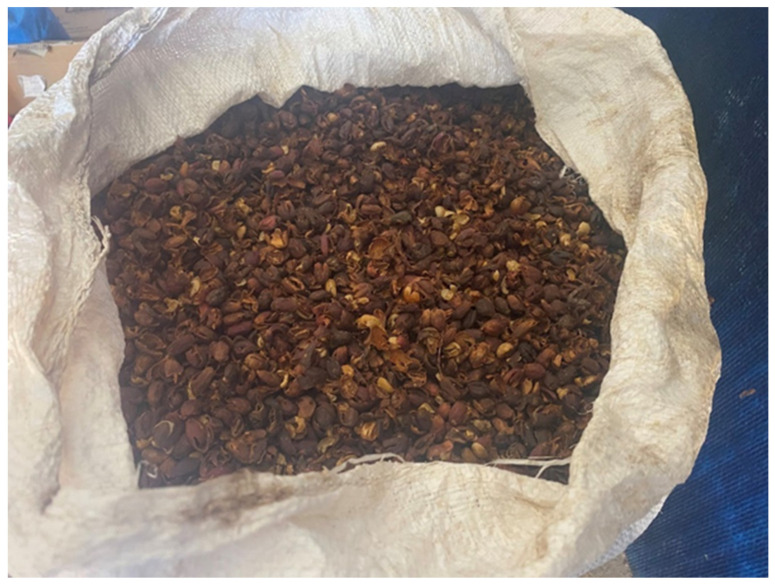
Coffee cherry husks obtained after dry processing.

**Figure 7 molecules-27-08435-f007:**
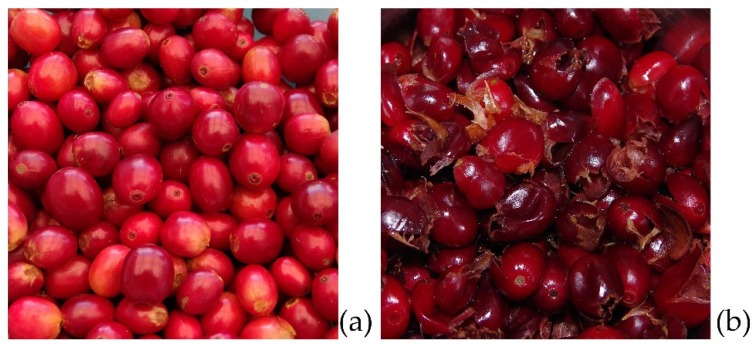
(**a**) Ripe coffee cherries before processing; (**b**) coffee cherry pulp obtained after wet processing.

**Figure 8 molecules-27-08435-f008:**
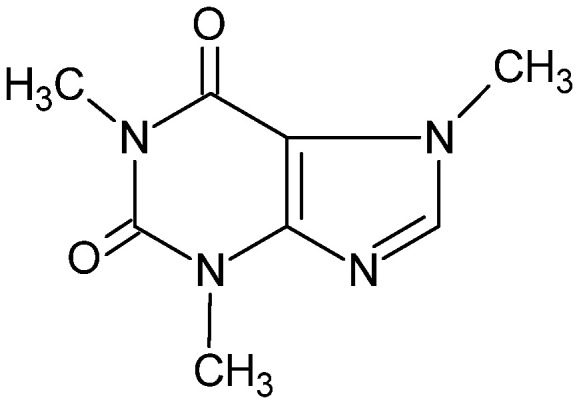
Chemical structure of caffeine.

**Figure 9 molecules-27-08435-f009:**
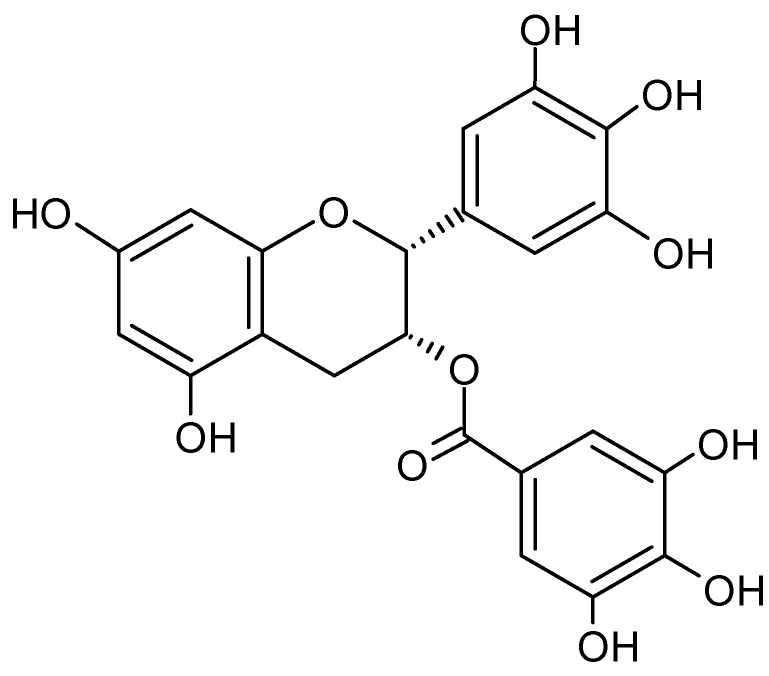
Chemical structure of epigallocatechin gallate (EGCG).

**Figure 10 molecules-27-08435-f010:**
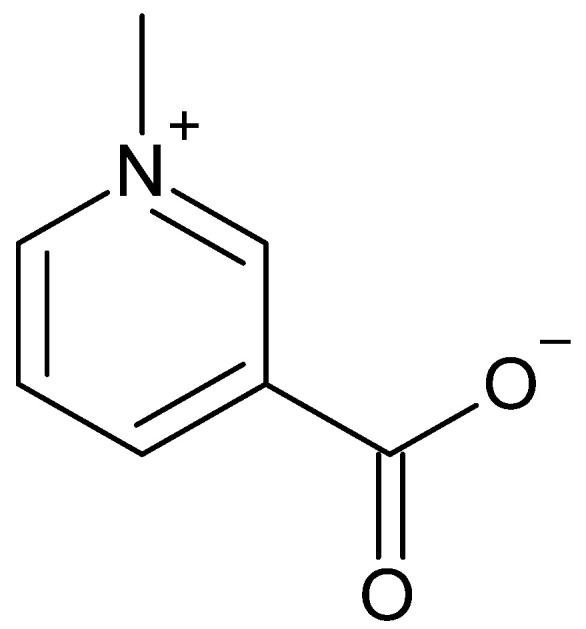
Chemical structure of trigonelline.

**Table 1 molecules-27-08435-t001:** Caffeine amount of coffee cherry pulp/husk.

*Coffea* Species	Sample Type (Sample Origin)	Converted Caffeine Content [g/kg]	Sample Preparation	Analytical Method	Source
*C. arabica*	Dried coffee husk (Nicaragua, Panama)	2.0 4.6 5.3 6.5 3.5 6.4 1.4	3 g in 100 mL distilled water at 95 °C for 5 min	HPLC/UV	[35]
*C. arabica*	Dried cascara pulp of coffee cherries, which were kept in a bag for 12 h before pulping	1.4	65.5 g in 1000 mL water at 90 °C for 6.5 min	Spectrophotometry	[1]
	Pulp soaked in water for 12 h	1.8			
	Pulp left in a basket for 12 h	2.5			
	Pulp/husk of dry processed coffee	3.7			
	pulp of wet processed coffee (Indonesia)	4.5			
*C. arabica*	Air-dried coffee husk (Mexico, India)	1.3 2.2	Coffee sample was ground to a fine powder; 20–30 mg extracted with 50% ethanol/water (7 mL) by vortexing the contents for 10 min followed by centrifugation; supernatant was transferred; the extraction was repeated 5 times; 5 mL-aliquots were freeze dried and reconstituted in 500 μL of methanol/0.1% formic acid	LC-MS	[36]
*C. canephora*	Air-dried coffee husk (Mexico, India)	0.9 1.0	Same as above	LC-MS	[36]
*C. arabica* (wet processed)	Coffee cherry pulp powder (Thailand)	0.0882	Chemical composition analysis: 1 g in 250 mL water at 90 °C for 20 min with 5.0 g of MgO	HPLC-DAD	[37]
			Conventional extraction: Distilled water was added to the dried coffee cherry pulp powder at a ratio of 1:20 (*w*/*v*) with a 15 min holding time at room temperature followed by heating in a water bath		
		0.0214	80 °C for 1 min		
		0.0186	80 °C for 3 min		
		0.0208	80 °C for 5 min		
		0.0186	90 °C for 1 min		
		0.0266	90 °C for 3 min		
		0.0254	90 °C for 5 min		
		0.0274	100 °C for 1 min		
		0.0204	100 °C for 3 min		
		0.0328	100 °C for 5 min		
*C. canephora* (wet processed)	Coffee cherry pulp powder (Thailand)	0.0137 0.0038 0.0052 0.0052 0.0050 0.0062 0.0064 0.0064 0.0070 0.0078	Same as above	HPLC-DAD	[37]
*C. arabica* (wet processed)	Cascara		Infusing 60.0 g of ground cascara in 600 mL deionized water at 93 °C for 5 min, then cooled in ice bath for 10 min, then filtered	HPLC	[38]
	(Brazil	3.2			
	El Salvador	2.1			
	Guatemala 1	3.5			
	Papua New Guinea	3.7			
	Guatemala 2)	2.1			
*C. arabica* (dry processed)	80% pulp + 20% parchment	6.2	Blanching: in water at 90 °C for 1 min 0.1 g in 10 mL boiling water; samples were conditioned in a Dubnoff bath at 100 °C for 10 min	HPLC	[39]
	80% pulp + 20% parchment blanched	5.2			
	28% pulp + 72% parchment	7.0			
	28% pulp + 72% parchment blanched (Brazil)	4.2			
*C. arabica* (wet processed)	Castillo	5.3	1.0 g in 20 mL of water for 15 min at 85 °C in a water bath and stirred on a magnetic plate for 10 min, then centrifuged	HPLC-DAD	[40]
	Colombia	7.2			
	Caturra	7.7			
	(Colombia)				

**Table 2 molecules-27-08435-t002:** Trigonelline amount of coffee cherry pulp/husk.

*Coffea* Species	Sample Type	Trigonelline Content [g/kg]	Sample Preparation	Analytical Method	Source
*C. arabica* (dry-processed)	80% pulp + 20% parchment 80%	5.4	Blanching: in water at 90 °C for 1 min; 0.1 g in 10 mL boiling water; samples were conditioned in a Dubnoff bath at 100 °C for 10 min	HPLC	[39]
	Pulp + 20% parchment blanched	2.9			
	28% pulp + 72% parchment	2.5			
	28% pulp + 72% parchment blanched	1.2			

**Table 3 molecules-27-08435-t003:** Caffeine and trigonelline amount of coffee cherry jam; below: caffeine content of common foods for comparison.

Size	Caffeine (mg)	Trigonelline (mg)
1 kg	399	567
1 glass (450 g)	180	255
1 large spoon (20 g, 1 portion)	8	11
Cola soft drink (255 mL)	29.5 [64]	n.a
Cup of coffee (255 mL)	74 [65]	n.a.
Sweet chocolate (28 g)	20 [66]	n.a.

**Table 4 molecules-27-08435-t004:** Daily actual caffeine intake by age group and safe limit values for single dose and acceptable daily intake (ADI).

Age Class	Safe Single Dose [32]	Acceptable Daily Intake ADI [32]	Caffeine Intake in mg/day (mg/kg bw)		Source
			Mean	P95	
Toddlers 12 to <36 months	n.a.	n.a.	5.9 (0.5)	27.3 (2.2)	VELS
Other children 3 to <10 years	3 mg/kg bw	5.7 mg/kg bw	13.5 (0.8)	47.4 (2.6)	VELS
Adolescents 10 to <18 years	3 mg/kg bw	5.7 mg/kg bw	59.4 (1.0)	208.1 (3.5)	NVS II
Adults 18 to <65 years	200 mg (3 mg/kg bw)	400 mg (5.7 mg/kg bw)	238.0 (3.2)	538.7 (7.3)	NVS II
Elderly 65 to <75 years	200 mg (3 mg/kg bw)	400 mg (5.7 mg/kg bw)	241.4 (3.2)	486.4 (6.3)	NVS II
Very elderly ≥75 years	200 mg (3 mg/kg bw)	400 mg (5.7 mg/kg bw)	208.2 (2.8)	397.9 (5.2)	NVS II
Population group					
Pregnant women	n.a.	200 mg			
Lactating women	200 mg	200 mg			

**Table 5 molecules-27-08435-t005:** Total amount of food that can be safely consumed at one time determined by using single-dose values.

Food	Caffeine		EGCG	Trigonelline	Safe Intake of Food ^a^
Juice	70–300 mg/L		n.a.	100–600 mg/L	43–10 mL/kg bw
Jam	399 mg/kg		n.a.	567 mg/kg	7.5 g/kg bw
Jelly	399 mg/kg		n.a	567 mg/kg	7.5 g/kg bw
Processed puree Unprocessed puree	400 mg/kg 480 mg/kg		n.a.	540 mg/kg 650 mg/kg	7.5 g/kg bw 6.3 g/kg bw
Wheat bread 2.5% CCHP	100 mg/kg		n.a.	135 mg/kg	30 g/kg bw
Egyptian flatbread 5% CCHP 10% CCHP 15% CCHP	Authors Measured value	Mean value (this study)	n.a.	19 mg/125 g 37 mg/125 g 56 mg/125 g	3–14 pcs 1–7 pcs 1–4 pcs
	63 mg/125 g 125 mg/125 g 188 mg/125 g	14 mg/125 g 28 mg/125 g 41 mg/125 g			

^a^ Calculated food amount containing 3 mg of caffeine (safe single dose per kg bw).

## Data Availability

No new data were created or analyzed in this study. Data sharing is not applicable to this article.

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
