# Peer review of "Risk Assessment of Coffee Cherry (Cascara) Fruit Products for Flour Replacement and Other Alternative Food Uses"

_molecules, 2022, doi:10.3390/molecules27238435_

Round 1
Reviewer 1 Report
The topic is current, but the scientific contribution of the project has significant weaknesses in terms of processing, relevancy and clarity. There are som examples:
Table 1 is confusing without relevant content that is aimed on unit conversion that has weak scientific importance Table 2 is not structured appropriately Comparison of caffeine content in different portions has weak scientific contribution and it is presented in Figure 11 and also in Table 3 Basic calculation for caffeine and trigonelline content in 30 g portion presented down at page 14 and below Table 6 has low scientific contribution Table 4 – detail presentation of average daily intake of food commodities according to gender and age is out of scientific scope related to risk evaluation of product from coffee cascara Table 5 and 6 are not very well arranged giving impression that represent first draft version Figure 3 – reference is missing Authors concluded that consumption of novel food products from coffee cherry fruit products could be considered as a safe for children over 3 years of age. However, for the risk assessment, it would be important to draw attention to the inappropriateness of consumption for specific target groups and/or to emphasize the maximum amounts of food with regard to the maximum possible concentrations.Authors highlighted in the abstract relevancy of utilization of powdered coffee cherry as a flour replacement for improving problems with raw material resourses and economic situation caused by the war in Ukraine, that is very hypothetical and out of research that have been provided by literature study.
Author Response
The topic is current, but the scientific contribution of the project has significant weaknesses in terms of processing, relevancy and clarity. There are some examples:
Table 1 is confusing without relevant content that is aimed on unit conversion that has weak scientific importance
RESPONSE: Table 1 now contains only the converted data. The order and appearance of the columns has been changed.
Table 2 is not structured appropriately
RESPONSE: Table 2 now contains only the converted data. The order of the columns has been changed for better clarity.
Comparison of caffeine content in different portions has weak scientific contribution and it is presented in Figure 11 and also in Table 3
RESPONSE: Figure 11 was eliminated and partially integrated into Table 3.
Basic calculation for caffeine and trigonelline content in 30 g portion presented down at page 14 and below Table 6 has low scientific contribution
RESPONSE: The calculations were deleted.
Table 4 – detail presentation of average daily intake of food commodities according to gender and age is out of scientific scope related to risk evaluation of product from coffee cascara
RESPONSE: Table 4 has been removed. The important information contained has been placed in the text.
Table 5 and 6 are not very well arranged giving impression that represent first draft version
RESPONSE: Tables 5 and 6 were modified.
Figure 3 – reference is missing
RESPONSE: Reference was added.
Authors concluded that consumption of novel food products from coffee cherry fruit products could be considered as a safe for children over 3 years of age. However, for the risk assessment, it would be important to draw attention to the inappropriateness of consumption for specific target groups and/or to emphasize the maximum amounts of food with regard to the maximum possible concentrations.
RESPONSE: The Conclusion is now worded more cautiously and identifies people for whom consumption is contraindicated.
Authors highlighted in the abstract relevancy of utilization of powdered coffee cherry as a flour replacement for improving problems with raw material resources and economic situation caused by the war in Ukraine, that is very hypothetical and out of research that have been provided by literature study.
RESPONSE: The need for flour blends was elaborated by current literature in the Introduction.
Reviewer 2 Report
Introduction part is appropriate but a few things are needed for further improvements especially the study aims should be added. Update the references
Add some studies about the study with highlighting research gaps, which necessitated conducting this trial.
Materials and methods:
-this part describes very well by using suitable subheadings. However, it needs few modifications and details of selecting primers and amplification conditions in the revised version to enhance
Author Response
Introduction part is appropriate but a few things are needed for further improvements especially the study aims should be added. Update the references
RESPONSE: The aim of the review is formulated in the risk assessment of coffee cherry products due to bioactive substances so that harvest by-products can be processed into food. More current references are now included.
Add some studies about the study with highlighting research gaps, which necessitated conducting this trial.
RESPONSE Since this is a review, we were able to refer to existing information and put it in a different context. Literature values are sufficient for risk assessment and derivation of safe consumption levels. No trial was needed to generate new data.
Materials and methods:
-this part describes very well by using suitable subheadings. However, it needs few modifications and details of selecting primers and amplification conditions in the revised version to enhance
RESPONSE: Presumably, this comment does not relate to the present paper, as the methods section has no subheadings and no work was done with primers and amplification methods.
Reviewer 3 Report
The manuscript “molecules-1966864-peer-review-v1” mainly develop “Risk assessment of coffee cherry (cascara) fruit products for flour replacement and other alternative food uses”. The review is a good starting point, starting with the by-products, however some more improvements are needed.
1. The key words in the article are not concise enough and the article is not coherent enough.
2. The impact on adolescents is not clear enough, whether it can replace the value of wheat in the development of adolescents.
3. The benefits of caffeine as a substitute for flour do not seem clear enough, and a small amount of use may not be harmful to humans, but can the value of wheat itself be replaced, and the cost of replacement? It is not clear enough. And only a single dose to determine whether it is a little less comprehensive.
4. Fresh coffee cherry pulp must be processed directly after harvest because the fermentation process starts quickly, does it prove difficult and costly to make jelly to ensure uniformity of taste?The authors believe that the triangle Au synthesizes particles extracellularly by MB proteins excreted from dead E. coli. Please supplement the control experiment that Au ions synthesize triangle Au in the medium in which MB protein is present.
5. More references from recent years should be added.
6. This paper is more like a poorly written lab report with various grammatical and lexical errors, so the authors should revise it carefully.
Author Response
The manuscript “molecules-1966864-peer-review-v1” mainly develop “Risk assessment of coffee cherry (cascara) fruit products for flour replacement and other alternative food uses”. The review is a good starting point, starting with the by-products, however some more improvements are needed.
- The key words in the article are not concise enough and the article is not coherent enough.
RESPONSE: Keywords have been adjusted where possible.
- The impact on adolescents is not clear enough, whether it can replace the value of wheat in the development of adolescents.
RESPONSE: Consumption scenarios for adolescents were exemplarily added with regard to limit values. In addition, the chapter on flour goes into more detail on nutritional and physiological aspects.
- The benefits of caffeine as a substitute for flour do not seem clear enough, and a small amount of use may not be harmful to humans, but can the value of wheat itself be replaced, and the cost of replacement? It is not clear enough. And only a single dose to determine whether it is a little less comprehensive.
RESPONSE: The nutritional benefits as well as the (socio-)economic and health effects of the flour substitute are now more detailed in the chapter on flour. The topic of costs is also discussed.
- Fresh coffee cherry pulp must be processed directly after harvest because the fermentation process starts quickly, does it prove difficult and costly to make jelly to ensure uniformity of taste?
RESPONSE: The practical procedure is now briefly outlined in the jelly part. There are defined work steps which ideally lead to a standardized end product. Only on a large scale with different batches will it become clear whether there is a uniformity of taste or whether there are natural variations in the end product.
The authors believe that the triangle Au synthesizes particles extracellularly by MB proteins excreted from dead E. coli. Please supplement the control experiment that Au ions synthesize triangle Au in the medium in which MB protein is present.
RESPONSE: The content of this comment cannot be related to the present paper. Perhaps a copy/paste mistake?
- More references from recent years should be added.
RESPONSE: Current references were added.
- This paper is more like a poorly written lab report with various grammatical and lexical errors, so the authors should revise it carefully.
RESPONSE: A careful revision was conducted as requested. We thank the reviewer for the various remarks and hope that the paper is now suitable to the high standards of the journal.
Reviewer 4 Report
I read the Review proposal “Risk Assessment of Coffee Cherry (Cascara) Fruit Products for 2 Flour Replacement and Other Alternative Food Uses” (molecules-1966864). The manuscript idea is interesting, however some text is over explained and it is difficult to read.
Some points:
Please reduce the amount of information in Introduction section, there are 6 pages and many information is not related to the article contain. In this section it would be desirable to include the proximal analysis of some coffee cherry cascara in order to have a first contact of possible applications.
Is necessary to include a materials and methods section in a review?
In Table 1, I would delete “Caffeine content (original unit)” column. Table 1 is discussed using “Caffeine units (g/kg)”. In my opinion, one column is enough.
Line 195, what about the analytical protocol employed? HPLC differs from UV-VIS?
Section 3.1.2. Epigallocatechin gallate. This section in my opinion is not necessary, there is a lot of information and at the end of the section it is concluded that its contribution is not significant.
Table 3. The term “Coffee cherry jam” is contains in Figure caption and in all the rows.
Section 3.3.2. Exposure to Epigallocatechin gallate is a discussion of a section non-significant.
Line 575. There are any information in the review to support the following conclusion: ”Another advantage is given with the possible positive effects on human health…” . The paragraph can be omitted.
Please check reference style (Reference 36, 40, 41, 44).
In general, the references are necessary but many of them have more than 10 years published.
Author Response
I read the Review proposal “Risk Assessment of Coffee Cherry (Cascara) Fruit Products for 2 Flour Replacement and Other Alternative Food Uses” (molecules-1966864). The manuscript idea is interesting; however, some text is over explained and it is difficult to read.
Some points:
Please reduce the amount of information in Introduction section, there are 6 pages and many information is not related to the article contain. In this section it would be desirable to include the proximal analysis of some coffee cherry cascara in order to have a first contact of possible applications.
RESPONSE: The Introduction has been shortened and the harvesting methods are now explained in the Results section.
Is necessary to include a materials and methods section in a review?
RESPONSE: Probably it is not strictly necessary, but we would like to present the review strategy.
In Table 1, I would delete “Caffeine content (original unit)” column. Table 1 is discussed using “Caffeine units (g/kg)”. In my opinion, one column is enough.
RESPONSE: Table 1 now contains only the converted data. The order and appearance of the columns has been changed.
Line 195, what about the analytical protocol employed? HPLC differs from UV-VIS?
RESPONSE: We re-checked the sentence, but were unable to confirm the claim of the reviewer. HPLC is often combined with a UV-VIS detector.
Section 3.1.2. Epigallocatechin gallate. This section in my opinion is not necessary, there is a lot of information and at the end of the section it is concluded that its contribution is not significant.
RESPONSE: We can understand the point. The part has been slightly shortened. Since epigallocatechin gallate is a bioactive substance that is often studied in connection with coffee harvest by-products, it should at least be explained and discussed.
Table 3. The term “Coffee cherry jam” is contains in Figure caption and in all the rows.
RESPONSE: The term was deleted in the rows as requested.
Section 3.3.2. Exposure to Epigallocatechin gallate is a discussion of a section non-significant.
RESPONSE: The EGCG part was slightly shortened. Further reduction would deprive important basics about EGCG. The risk assessment is as concise as possible.
Line 575. There are any information in the review to support the following conclusion: ”Another advantage is given with the possible positive effects on human health…” . The paragraph can be omitted.
RESPONSE: We agree and have deleted the paragraph.
Please check reference style (Reference 36, 40, 41, 44).
RESPONSE: The references were checked and revised.
In general, the references are necessary but many of them have more than 10 years published.
RESPONSE: The authors wanted to be as comprehensive as possible. The topic is rather restricted so that we cannot only rely on references of the last 10 years. We have included some further new references published during the time of the peer review process.
Round 2
Reviewer 3 Report
Accept
Reviewer 4 Report
The authors answer all the suggestions proposed. In my opinion the manuscript can be accepted.